# Preventing Drift through Continued Co-Design with a First Nations Community: Refining the Prototype of a Tiered FASD Assessment

**DOI:** 10.3390/ijerph191811226

**Published:** 2022-09-07

**Authors:** Luke Miller, Dianne C. Shanley, Marjad Page, Heidi Webster, Wei Liu, Natasha Reid, Doug Shelton, Karen West, Joan Marshall, Erinn Hawkins

**Affiliations:** 1Gold Coast Hospital and Health Service, Gold Coast 4215, Australia; 2Menzies Health Institute Queensland, Griffith University, Gold Coast 4222, Australia; 3School of Applied Psychology, Griffith University, Gold Coast 4222, Australia; 4Child Health Research Centre, University of Queensland, Brisbane 4101, Australia; 5Aboriginal and Torres Strait Islander Community, Mount Isa 4825, Australia

**Keywords:** neurodevelopmental assessment, first nations peoples, co-design, fetal alcohol spectrum disorder, primary health care

## Abstract

As part of the broader Yapatjarrathati project, 47 remote health providers and community members attended a two-day workshop presenting a prototype of a culturally-safe, tiered neurodevelopmental assessment that can identify fetal alcohol spectrum disorder (FASD) in primary healthcare. The workshop provided a forum for broad community feedback on the tiered assessment process, which was initially co-designed with a smaller number of key First Nations community stakeholders. Improvement in self-reported attendee knowledge, confidence, and perceived competence in the neurodevelopmental assessment process was found post-workshop, assessed through self-report questionnaires. Narrative analysis described attendee experiences and learnings (extracted from the workshop transcript), and workshop facilitator experiences and learnings (extracted from self-reflections). Narrative analysis of the workshop transcript highlighted a collective sense of compassion for those who use alcohol to cope with intergenerational trauma, but exhaustion at the cyclical nature of FASD. There was a strong desire for a shared responsibility for First Nations children and families and a more prominent role for Aboriginal Health Workers in the assessment process. Narrative analysis from workshop facilitator reflections highlighted learnings about community expertise, the inadvertent application of dominant cultural approaches throughout facilitation, and that greater emphasis on the First Nation’s worldview and connection to the community was important for the assessment process to be maintained long-term. This study emphasised the benefit of *continued* co-design to ensure health implementation strategies match the needs of the community.

## 1. Introduction

Accessible and culturally-safe neurodevelopmental assessments identifying fetal alcohol spectrum disorder (FASD) are needed for First Nations Peoples living in remote Australian communities. While Australia does not currently have national FASD prevalence data [1], the global prevalence rate is estimated to be 7.7% [2]. These rates can be as high as 19.4% for First Nations children living in remote Australian communities [3]. Children experiencing FASD face a unique set of challenges throughout their childhood and are more likely to experience further mental health problems, and loss of educational and vocational opportunities into young adulthood [4,5,6]. Assessing and identifying FASD is an important phase in assisting people to gain access to effective healthcare services. However, some neurodevelopmental assessments do not sufficiently investigate FASD as a potential diagnosis [7] and health professionals’ knowledge about diagnosing and supporting FASD is often limited, particularly in remote communities [8]. 

Neurodevelopmental assessments are typically completed by specialised medical and allied health practitioners (e.g., paediatricians, psychologists, speech pathologists, occupational therapists) in multidisciplinary clinics—clinics and clinicians that are scarcely, if at all, available in remote communities [9]. Primary health practitioners are a more stable workforce in these communities, with Aboriginal health practitioners/workers the most stable available workforce [10,11]. Aboriginal health practitioners provide direct support to patients and assist other health providers to better understand and respond to First Nations Peoples’ needs.

Capitalising on a more stable and culturally-safe primary health care workforce may help increase access to services in remote Australian communities, while also facilitating early detection and monitoring of neurodevelopmental delays and prenatal alcohol exposure. Early identification and support for neurodevelopmental delays and disorders can occur as part of routine health checks [12], but current health data suggests that 70% of Australian First Nations Peoples between the ages of 0 and 14 years do not attend health checks [13]. This data highlights the ongoing inequities in healthcare for First Nations Peoples despite efforts to close the gap [14].

A major contributor to this inequity is the lasting impact of colonisation on Australia’s First Nations Peoples. Acknowledging the atrocities of colonisation and dispossession, and the lasting influence it continues to have on First Nations Peoples is critical to improving health and wellbeing [15,16]. Health providers can play a key role in supporting this acknowledgement and making progressive steps forward in closing the gap. First, understanding that social and cultural determinants of health and neurodevelopment, such as intergenerational trauma and loss of cultural connection, make it difficult to escape intergenerational cycles of adverse health outcomes [17,18,19,20]. Second, understanding that adverse health outcomes are also perpetuated by structural inequalities and systemic racism, resulting in feelings of fear and distrust, ongoing oppression, and discrimination against First Nations Peoples [20]. Finally, by valuing cultural identity as an important determinant of health and wellbeing [21]. 

### 1.1. The Yapatjarrathati Project

The Yapatjarrathati project aimed to address some of these health inequities by utilising co-design to collaboratively redesign a more culturally-safe tiered neurodevelopmental assessment process that can identify FASD [22]. The process began by acknowledging that three-quarters of the research team and clinical specialists were not First Nations, were not from the local community, and therefore brought with them a Western cultural and medical bias. To ensure the voices of First Nations Peoples were valued and prioritised, there was a period of informal relationship building and learning with key Elders and community stakeholders in the initial phase of the co-design process. Co-design consultations were then expanded to include local health providers and the community at large. The inclusion of arts-based participatory research methods summarised learnings from these consultations and a prototype of a tiered neurodevelopmental assessment process was developed. 

Two broad community co-design workshops were planned before implementing the prototype of the neurodevelopmental assessment process to ensure adaptation to the specific needs and concerns of the local community (e.g., community history, presenting concerns of their children, workforce capacity issues). The first broad community co-design workshop focused on discussions of the causes and consequences of FASD and how to assess at risk children locally [15]. The team gained valuable learnings about the community’s view on the socio-cultural factors contributing to the prevalence of FASD in this community and how to best proceed with implementing services aimed at identifying and supporting children with FASD and their families locally [15]. The acknowledgment and inclusion of First Nations Peoples’ knowledge and practices and the ongoing use of a collaborative, bottom-up approach were key findings from this workshop [15]. In line with these findings, the assessment prototype was further refined and then presented back to the broader community in a second workshop four months later. 

### 1.2. Content of the Workshop and Study Objectives

The current study documents the ongoing co-design process with this community, focusing on the second broad community consultation in which the prototype of the tiered neurodevelopmental assessment was presented. The assessment prototype reshaped current best practices in FASD assessment, couching all elements of a comprehensive neurodevelopmental assessment within a Dreamtime story. A Dreamtime Story is a cultural practice used for teaching and remembering culturally important information [23]. The tiered approach follows the Australian guide to the diagnosis of FASD [24], but divides the assessment process into six manageable parts that can be completed in primary healthcare settings by different combinations of health providers depending on clinical need and availability [15,22]. The Dreamtime story is introduced in Tier 1 to explain the assessment journey and obtain informed consent. Information about the child’s history is obtained in Tier 2 using a culturally sensitive developmental interview. In Tier 3, an Australian adaptation of the Rapid Neuro-Developmental Assessment is administered [25,26,27,28], which involves rapidly assessing hearing, vision and seven broad neurodevelopmental domains (e.g., motor, cognition). Tier 4 involves obtaining information from other people in the child’s life and environment, including parent- and teacher-report of social, emotional, behavioural, adaptive, and academic functioning. Tier 5 ensures the family receives feedback and support consistently across health providers. Tier 6 provides further specialist assessment, as clinically indicated. 

This study aimed to quantitatively assess gains in workshop attendee knowledge about FASD and neurodevelopmental assessments, confidence in the assessment prototype, and perceived competence to implement the assessment post workshop. We also aimed to gain qualitative feedback on the implementation of the assessment prototype within the local community. Given the large proportion of non-First Nations team members and workshop facilitators, there is potential for program drift due to the inherent biases of theses collaborators. Qualitative feedback of the facilitator experience and how information learned from the attendees could guide the direction of the larger project was also assessed. 

## 2. Materials and Methods

### 2.1. Research Design

A two-phase multi-method approach was used. The first phase involved concurrent collection of quantitative pre-post attendee surveys and qualitative attendee discussions throughout the workshop [27]. The second phase involved the qualitative collection of facilitator data post-workshop. Qualitative and quantitative components of the analysis were weighted equally in keeping with an ‘all teach-all learn’ approach [22,29]. Data were first analysed separately and integrated in the interpretation to allow for greater understanding of the results and application of the workshops. 

### 2.2. Participants

*Attendees:* The workshop was targeted to participants who attended the first community consultation workshop [15] and other stakeholders recruited by local Elders and respected community members. An advertisement about the workshop was emailed to the 92 participants who registered for the first community consultation [15]. Flyers were also hand delivered to community members and stakeholders by local Elders. Forty-nine members of the community and employees in the region’s health sector registered to attend the workshop with 47 attending the workshop (41 female). All attendees were eligible to participate in the study. Demographics for the attendees are presented in Table 1. The mean age of attendees who answered this question (*n* = 38) was 35.97 (SD = 11.28) years. Approximately 40% of attendees identified as Aboriginal or Torres Strait Islander Peoples, representing 13 different nation groups. Attendees had various roles in the community, with Aboriginal health workers/practitioners (*n* = 7), and allied health practitioners (*n* = 18) being the most common. Attendee work experience ranged from 0 months to 28 years. Forty-two attendees completed the pre-workshop questionnaire, and twenty-five (24 female) also completed the post-workshop questionnaire. Five attendees completed the post-workshop questionnaire only. There were no significant differences between age, experience, pre-workshop knowledge, pre-workshop confidence, and pre-workshop perceived competence when comparing participants who completed both questionnaires to those who only completed the pre-workshop questionnaire. Of the people who completed the pre-workshop questionnaire, 6 identified as Aboriginal or Torres Strait Islander Peoples and only 2 attendees completing both questionnaires identified as Aboriginal or Torres Strait Islander Peoples.

*Facilitators:* There were four clinician/researcher facilitators (3 female, 1 male). All four facilitators were clinical specialists and researchers on the Yapatjarrathati project team. Two facilitators were clinical psychologists and one a community paediatrician. These facilitators were not First Nations people, and not from the local community. They had specialist knowledge in FASD, neurodevelopment, and/or health system change. The fourth facilitator was a First Nations general medical practitioner from the local community. This facilitator was the author of the Dreamtime story and had specialist knowledge in FASD, primary care, and community health. 

### 2.3. Materials and Methods

Attendee knowledge was assessed using seven multiple-choice questions that asked about the assessment process (e.g., ‘what is the easiest way to get information from the school for the assessment’), and FASD criteria (e.g., ‘How many brain domains need a severe score to meet criteria for an FASD diagnosis’). Correct responses were scored as ‘1’, and incorrect responses were scored as ‘0’. Missing data were treated as incorrect. The seven items were then summed to give a total score, with higher scores indicating greater knowledge. 

Attendee confidence in completing the prototyped assessment process was measured using 13 questions on a 5-point Likert scale (1 = *very confident*, 5 = *not confident*). This included using rapid assessment tools (e.g., ‘completing the rapid neurodevelopmental assessment [RNDA]’), communicating with other professionals (e.g., ‘following up with GPs to complete their checklist’), and helping families (e.g., ‘suggesting strategies that might help children and families’). The items were summed and divided by the total number of items to provide a mean score. A lower score indicated higher confidence. Internal consistency was very good with a Cronbach’s α of 0.94 (pre-workshop) and 0.94 (post-workshop).

Attendee perception of competence was assessed using 22 questions on a 5-point Likert scale (1 = *strongly agree*, 5 = *strongly disagree*). These questions included topics such as assessment of FASD (e.g., ‘my knowledge of how to assess FASD is up to date’), empathetic communication with clients (e.g., ‘I know how to communicate with individuals with FASD’), and improving outcomes for clients (e.g., ‘I am able to provide effective strategies to support individuals with FASD’). The items were summed and divided by the total number of items, to provide a mean score. A lower score indicated higher perceived competence. Internal consistency was very good with a Cronbach’s α of 0.96 (pre-workshop) and 0.95 (post-workshop).

### 2.4. Procedure

*Workshop*. Consent to conduct this research was obtained from the regional committee that represents the Traditional Owners of the land. Ethics approval was granted by Children’s Health Queensland Hospital and Health Service Human Research Ethics Committee (HREC/18/QRCH/127) and Griffith University Human Research Ethics Committee (GU Ref No: 2018/747). The workshop was conducted over two days in community in north-west Queensland, Australia. The first day of the workshop focused on Tier 3 of the neurodevelopmental assessment prototype, reviewing the administration and scoring of the Australian adapted RNDA [25,26,27,28]. This involved demonstration of the assessment measure, hands-on practice, and scoring of case samples. Case samples were hypothetical cases derived from different presentations seen in clinical practice by the facilitators. The second day of the workshop provided an overview of the entire neurodevelopmental assessment process. There was time during each component of the workshop for ‘yarning’, an Australian First Nations conversational process of telling and sharing stories and information [30]. These conversations were purposely non-directive, allowing the yarn to develop without direct questioning. The use of yarning, an Indigenous research methodology, was important to maintaining collaborative and respectful relationships between the research team and community [23], and was consistent with the larger project’s ‘all teach, all learn’ philosophy [22,29]. 

*Attendee qualitative data*. With participant permission, the workshop was video recorded and group discussions were transcribed by one of the authors who did not attend or facilitate the workshop (NR). 

*Attendee quantitative data*. Attendees completed the pre-workshop questionnaire at the beginning of the first day and at the end of the second day of the workshop. The questionnaire included general demographic information, including age, ethnic background, current position, and years of experience in that position. The questionnaire posed questions related to the attendee’s knowledge and confidence about the tiered assessment, and attendees perceived competence in assessing and supporting children with FASD. The questionnaire was developed to match the content discussed during the workshop. 

*Facilitator qualitative data*. To document reflexivity and the perceived influence of community feedback on the facilitators who are also members of the broader Yapatjarrathati project team, a series of six open-ended questions encouraging self-reflection were sent to workshop facilitators 18 months post-workshop (e.g., ‘What did you learn from the workshop?’, ‘How did your own worldview and biases influence how the workshop was run?’). Workshop facilitators were asked to reflect on the delivery of the workshop and whether and how subsequent implementation of the assessment prototype had shifted based on the community feedback. The six questions along with the original transcripts from the workshop were provided to the facilitators by email. Each facilitator was asked to review the workshop transcripts, consider participant comments and the implementation of the assessment process over the past 18 months, and provide a written response to the questions. 

### 2.5. Statistical Analysis

*Quantitative*. Quantitative analysis was conducted using IBM SPSS Statistics for Windows version 27 (IBM Corp., Armonk, NY, USA). Data was screened for outliers and missing data. There was 5% of missing data across both pre-and post-workshop questionnaires. Little’s MCAR test indicated that missing data was missing completely at random. However, one participant did not answer any perceived confidence items on the post-workshop questionnaire, while another participant answered only 1/22 of perceived competence items on the pre-workshop questionnaire. These participants were removed from the sample. One participant did not complete any knowledge items on the post-workshop questionnaire, which resulted in a total score of 0 for this construct. Analyses were performed with and without this participant. The removal of this participant did not alter the results; thus, the participant was retained. The final sample for data analyses was 23. As the assumption of normality was met, paired *t*-tests were used to determine if there was a difference between pre-and post-workshop for measures of participant knowledge, confidence, and perceived competence. Cohen’s *d* was used to determine effect sizes. 

*Qualitative.* The transcript from the workshop and written reflexive responses from the facilitators were analysed using a narrative analysis approach as described by Emden (1998) [31]. Specifically, this involved: (1) Reading the full transcript several times within an extended time frame (several weeks) to grasp the content; (2) Deleting all the questions and comments made by the workshop facilitators; (3) Deleting all words that detract from the key ideas of each sentence or group sentences; (4) Re-reading the remaining text; (5) Repeating steps 3 and 4 several times until satisfied that all key ideas are retained and extraneous content is eliminated; (6) Identifying fragments that constitute key themes; (7) Moving themes together to create one coherent core story or series of core stories; (8) Returning the core story to the respondents to check for accuracy and to determine if they want to correct or develop any part of the narrative. A narrative analysis considers narrative accounts as units rather than splitting comments into categories [32]. This was considered more culturally appropriate for this study [15]. Steps 1–3 for the narrative analysis for the workshop transcript were completed by author NR, with all remaining steps and steps 1–8 for narrative analysis for written reflexive responses completed by the first author who was completing a post-graduate research project. Author EH reviewed steps 4–8. 

## 3. Results

### 3.1. Attendee Knowledge, Confidence and Perceived Confidence in the Assessment Prototype

Quantitative analyses indicated that attendee knowledge increased post-workshop (M = 5.22, SD = 1.57) compared to pre-workshop (M = 3.65, SD = 1.77), *t*(22) = 5.59, *p* < 0.001 with a large effect size present (*d* = 1.34). Attendee confidence improved from pre-workshop (M = 2.85, SD = 0.88) to post-workshop (M = 2.23, SD = 0.71), *t*(22) = 4.22, *p* < 0.001 with a medium effect size (*d* = 0.71). Attendee’s perception of their competence improved from pre-workshop (M = 3.23, SD = 0.78) to post workshop (M = 2.21, SD = 0.42), *t*(22) = 6.96, *p* < 0.001, with a medium effect size (*d* = 0.75).

### 3.2. Narrative Analysis of Attendee Contributions

A broad storyline and series of key themes emerged through qualitative analysis of workshop attendee discussion: (1) Ensuring awareness of the ongoing impacts of colonisation, intergenerational trauma and systemic racism is acknowledged by practitioners engaged with First Nations Peoples’ health; (2) A collective sense of compassion for those who use alcohol to cope, but exhaustion due to the cyclical and enduring impact of alcohol use and FASD on the community; (3) Frustration at the Westernised approach of professionals working within silos instead of embracing a collaborative, holistic, community-centred approach; and (4) The importance of ensuring a prominent role for Aboriginal health workers in the assessment process, given their standing in the community and their knowledge of connection as integral to a stronger system. 

**Theme 1: Ensuring awareness of the ongoing impacts of colonisation, intergenerational trauma, and systemic racism is acknowledged by practitioners engaged with First Nations Peoples’ health.** A common and powerful message from attendees highlighted the ongoing impacts that colonisation and intergenerational trauma continued to have on the community and First Nations Peoples’ health and wellbeing. As one attendee stated,


*A lot of our mob give up and we talk about why warriors lie down and die. Well, it’s all the stress, trauma, and those things that impact on our mob’s everyday life.*


Another attendee highlighted the impact of racism and how it erodes trust and sharing health information.


*It [racism] impacts on our mob and us as workers. It is a separate entity. It is the most challenging barrier to get into any space. To get our mob to be able to talk.*


The importance and need for health care practitioners to have awareness and understanding of these issues before engaging in work in the community was also raised by an attendee, 


*No one should be working with our Aboriginal kids or our Aboriginal health unless you have taken into account the history, our history. Our dispossession of land, our loss of culture, and how that still affects a lot of our Elders and how that is passed down. The new word is inter-generational trauma and it’s still there.*


The impact of colonisation is still felt today by the First Nations community, and there must be practitioner acknowledgment of the prominent role this plays in First Nations Peoples’ ill health for real change to be realised. 

**Theme 2: A collective sense of compassion for those who use alcohol to cope, but exhaustion due to the cyclical and enduring impact of alcohol use and FASD on the community.** Several comments highlighted the cyclical and enduring nature of alcohol use within families as a means of coping with the ongoing impact of colonisation. For instance, one attendee presented this double-edged sword,


*And when you look at our mob today, with all that is going on in our lives and everything that impacts on them, the only way they can find any salvation or any hope [is] through alcohol usage. Unfortunately, that has impacted on our children, their children so we talk about the four generations of alcohol abuse and the impact on our mob.*


Another attendee commented,


*The other day I saw a young girl who was going (sic) dealing with sorry business and she said to me ‘I’ve cut down drinking, but because I can’t heal the spiritual and emotional trauma, I want to drink’.*


The embedding of alcohol use within the community and the consequences of this was also highlighted by an attendee:


*Our mob are starting to drink at 12. So, when you got that kid with FASD consuming alcohol themselves. We are really faced with the ugliest demon out there.*


**Theme 3: Frustration at the Westernised approach of professionals working within silos instead of taking shared responsibility with a community-centred approach.** This theme highlighted the collective frustration that service providers tended to work exclusively within their own discipline or organisations, with little to no collaboration between professionals and community members. Attendees spoke cogently about their experience with this and the desire for change in the way organisations operate towards working on the bigger problem together with one attendee stating,


*We need to not only be together in this room, but to be together outside this bloody door. Because the minute we walk out this door. Everybody is in their own little silo. In their own little predicament. No one wants to share, and we are all here to help. But it is not seen outside these doors. We can sit in here and talk, but out there you become your individual organisations, which is shit, because we are working for the same people and the same journey and for this to be successful and to work we need each other. *


Another participant expressed a similar view,


*We shouldn’t be working in little pockets. We should be out helping one another…We shouldn’t be saying that’s not our problem. That’s what we have been getting for the last 20 years…That’s the only way we are going to make a difference, is everyone working together.*


A change in the approach to making change is desired, with members of the community yearning for more collaboration, fighting the problem together instead of working in isolation. 

**Theme 4: The importance of ensuring a prominent role for Aboriginal health workers in the assessment process, given their standing in the community and their knowledge of connection as being integral to a stronger system.** Enhancing the connection with the community was highlighted by attendees as a key component to success moving forward, 


*Until that kid or that family trusts you enough, they won’t do any of this here. For anyone working with our mob. That relationship—don’t give up. If you are there for the right reason and our mob know that, our mob will be a bit more open to let them into their space.*


The importance of maintaining a strong connection with the community is already understood and promoted by Aboriginal health workers with one attendee succinctly stating,


*The [Aboriginal] health workers today keep on emphasising how important that connection is.*


Many attendees saw the opportunity for Aboriginal health workers to play a prominent role in the assessment process given their cultural expertise and knowledge of the community. This has not always been the case in other services, as one attendee noted, 


*[Aboriginal] health workers being a crucial component in this journey. But as a health worker we are always fighting the system because there is always that conflict, the health workers are always judged, rather than us being the expert, because we live in this community, we know our mob, we know how to communicate with our mob. We need to be recognised for the expertise that we have…*


Although Aboriginal health workers have not always been recognised and valued in the health system, there was strong community agreement about the critical need for them in this assessment process to ensure cultural safety, and to uphold the value of holistic care and connection.

### 3.3. Narrative Analysis of Workshop Facilitator Reflections

Analysis of the workshop facilitator reflections revealed key themes and a storyline of their learnings: (1) Recognition that despite good intentions, there was an inadvertent drift back to a dominant culture approach, by using a strategy of breaking things down into parts and focusing on knowledge acquisition at the expense of group discussion; (2) a renewed commitment from all that the community were the true experts to drive the desired change; and (3) a greater valuing of cultural knowledge and connection by making the Dreamtime story the centrepiece of implementation.

**Theme 1: Inadvertent drift back to a dominant culture approach, by breaking things down and focusing on knowledge acquisition at the expense of group discussion.** Despite months of broad community consultation and careful attempts to listen deeply to the needs of Elders and community stakeholders throughout the co-design journey, it was still easy for workshop facilitators to bring in their own biases and worldview to the workshop, with one facilitator highlighting:


*My background led me to focus on presenting the details and demonstrations of the assessment, provide the reasoning and logic behind decision-making and present case stories showing the process.*


But identifying they would do things differently in future, as indicated by this reflection:


*The first day would have more about community yarning, sharing of the impacts of history and culture and broader determinants of health, more time and space for participant feedback.*


Reflecting on how the facilitators drifted back to biases in line with their own culture and professional training allowed recognition and identification of changes to make moving forward with the project. 

**Theme 2: The community are the true experts to drive the desired change in their community.** It was clear that although the expert role was comfortable for many of the facilitators, there was a recommitment to trust in the expertise within the community, indicated by this facilitator,


*I quickly realised that I don’t have all the answers, that I’m not an expert in their community, and don’t have the grassroots knowledge of all the barriers/factors that impact.*



*I walked away from this workshop truly acknowledging that I didn’t know how to deliver FASD services in this community. I realised that this was okay, because the community knew—we just needed to listen.*


**Theme 3: There should be greater value on cultural knowledge and connection by making the Dreamtime story the centrepiece of implementation.** Each facilitator commented that the community feedback contributed to these two key changes in the implementation protocol. For instance, one of the facilitators noted,


*I would also be more focused on the Dreamtime story at more moments, showing everyone how the Dreamtime story brings everything we are talking about to life.*



*We have corrected this in our online training modules. The Dreamtime story comes first. Aboriginal health workers are also much more central to the training. Aboriginal health workers are part of the presentation/training team and encouraged to take on a larger role.*


This same facilitator also highlighted how community feedback helped shift the team away from thinking of the assessment as merely gathering information, but to supporting connection. 


*Our focus shifted from ‘assessing children’ to helping children and their families connect, and through that connection their health outcomes would improve, because when families feel connected, they start to thrive, from attending more appointments—to spending more quality time with the people who care about them.*


Another facilitator reflected on the importance of the connection between everyone involved:


*The implementation needs to be in parallel with the community, going on the journey together and not in segregation.*


## 4. Discussion

This study documented the outcomes of a workshop that introduced the prototype for a culturally-safe, tiered, neurodevelopmental assessment process to local health providers and community members in remote Australia. The tiered assessment was co-designed with key local First Nations stakeholders. The workshop provided space for the broader community to refine ideas and discuss implementation strategies so that the tiered assessment would be sustained over the long term and would truly suit community needs. The results indicated that the two-day workshop was effective in increasing workshop attendees’: knowledge about the prototyped assessment process and FASD, confidence in conducting the assessment including the RNDA, and perceptions of competence to complete the assessment. 

The most valuable lessons learned emerged from qualitative attendee feedback. Specifically, members of the community were frustrated with the Westernised approach to their problems, which reinforced feedback from the first workshop that the community wanted substantial involvement to ensure their needs were met [15]. Feedback from workshop attendees afforded the opportunity for reflexive practice by the workshop facilitators, who were also members of the larger research team working to implement and evaluate the assessment process in this community. As demonstrated in other studies, reflexive practice was important in this project to help facilitators recognise when and how their personal and professional background and values influenced the direction of the project [33,34]. This study showed that, even after more than a year of building relationships with the local community and working to value and uphold another worldview, facilitators from the dominant culture acknowledged inadvertently drifting back to their own worldview. Specifically, they noticed a drift back into an ‘expert’ role, breaking things down during the workshop and focusing on assessment tools and cases, rather than using the more holistic approach of the Dreamtime story, in which the same information was embedded. 

Consistencies in feedback across this and the first broad community consultation workshop included the impact of colonisation on alcohol use and FASD in the community, and the importance of collaboration and recognising First Nations Peoples’ knowledge [15]. However, feedback from the current workshop was more specific in recommending that, when implementing the assessment protocol, it must break down the current approach of working in ‘silos’ within the community. It must also make room for the Aboriginal health worker to take the lead in guiding the assessment process. 

As this study demonstrated, community feedback from the current workshop led the Yapatjarrathati project team to re-commit to ensuring that First Nations knowledge was at the forefront of service delivery. This meant that art and story guided all aspects of the model. Going forward, there was a shared responsibility among participating health providers and organisations to use the Dreamtime story to guide healthcare practice, and that Aboriginal health workers would be the most appropriate and effective health practitioners to lead the assessment journey.

The findings in this study confirm the need for *continuous* co-design and strong, genuine relationships with the community to prevent drift back to dominant cultural ways when implementing health services with and for First Nations communities. Continuous co-design and stakeholder engagement throughout the life of a co-designed project has offered checks and balances for ensuring outcomes were community- rather than expert-focused, demonstrating positive outcomes in other studies [35,36]. Brief and limited co-design sessions at the start of a project, or committees that involve community members, but do not engage in consistent co-design throughout the implementation stages, can lead to issues in achieving desired outcomes [37]. 

Just as co-design is important in redesigning services to better localize evidence-based practice to a specific community, sustainability will require quality improvement protocols with ongoing engagement from the community [38]. Ongoing community consultation workshops and informal communication with key stakeholders are important to ensure that community needs are heard and met. It is important that future consultation workshops focus on continuing to develop attendee knowledge, confidence, and competence in the model, but still allow for an opportunity for community feedback, as community needs can shift over time. In this project, feedback was the lynch pin to ensuring collaborative decision-making, guarding against a tendency to drift back to dominant cultural ways, and keeping First Nations Peoples’ cultural values as the cornerstone of service delivery.

### Limitations

A key component to effectively implementing the new assessment process is the attendee’s competence in executing the tiers correctly. Whilst the measure in the workshop provided a perception of attendees’ competence, this does not indicate the ability of the attendee to perform the tiers correctly. This limits the conclusions that can be drawn regarding the technical readiness of the attendees to implement the model. Further training and evaluation throughout the implementation process of the wider project is needed to ensure fidelity [22]. 

Additionally, measures of knowledge, confidence and perceived competence were completed immediately following the workshop without follow-up. Whilst this indicates understanding at the time, it does not demonstrate long-term learning, and a follow-up post-workshop may have been beneficial to assess any long-term learning. Monitoring these constructs throughout the implementation process is an important part of the evaluation. Another limitation of the study was the lack of male attendees involved in this workshop. However, many phases of the overall project [22] involved males within the community, which mitigated the risk of gender bias in the development of the overall assessment process [39,40].

## 5. Conclusions

This study highlighted the importance of ongoing workshops and collaboration when reshaping and implementing new health services that aim to meet the needs of a remote First Nations’ community. Results displayed effectiveness in increasing attendee knowledge, confidence, and perceived competence with the proposed assessment process but more importantly provided space for more discussion with the community as part of continuous co-design, helping to refine the new tiered, neurodevelopmental assessment process. The study shows the necessity of ongoing broad community feedback and monitoring of the co-design process through reflexive practice since there is a tendency to drift back into dominant culture or ‘expert’ roles when working with diverse cultural groups. As the project transitions to the next phases of implementation, the centrality of First Nations Peoples’ knowledge (demonstrated through the Dreamtime story) and having a high involvement of Aboriginal health workers will guide the project team.

## Figures and Tables

**Table 1 ijerph-19-11226-t001:** Workshop attendee demographics (*n* = 47).

	Frequency (%)
Identified as First Nations	
Aboriginal or Torres Strait Islander	19 (40.4)
Prefer not to answer	1 (2.1)
Gender	
Female	41 (87.2)
Male	6 (12.8)
Occupation	
Aboriginal health practitioner/worker	7 (14.9)
Allied health	18 (38.3)
Education professional	3 (6.4)
Child Health Nurse	5 (10.6)
Child Safety Officer	4 (8.5)
Student	1 (2.1)
Other	9 (19.1)
Sector	
Aboriginal Controlled Health Service	11 (23.4)
Non-government/not-for-profit	11 (23.4)
Queensland Health	11 (23.4)
Primary Health Networks	2 (4.3)
Department of Child Safety, Youth and Women	4 (8.5)
Education Queensland	1 (2.1)
Catholic/Private Education	4 (8.5)
Other	3 (6.4)

## Data Availability

Data is available on request from the corresponding author.

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
