# Peer review of "Preventing Drift through Continued Co-Design with a First Nations Community: Refining the Prototype of a Tiered FASD Assessment"

_ijerph, 2022, doi:10.3390/ijerph191811226_

Round 1

Reviewer 1 Report

This paper builds on the authors' prior work on the creation and implementation of a culturally grounded Tiered FASD assessment, specifically focusing on how this assessment is received by the broad health/service provider practitioner community. This is always an important step for community change. I also appreciate the emphasis on needing continual effort and iterative collaboration, and how co-creations should not be static. 

As this study builds on prior work, I understand that it doesn't make much sense to elaborate on parts that have been discussed elsewhere (like the assessment itself). However, my main comment is that I would like to see more detail about the context behind the conversations that provided the basis of the qualitative research. For example, in the workshop, how was reflection, knowledge, and experience shared? Were there prompts, or were things discussed after different subjects/lessons/lectures/etc? I think this is important in particular because although the authors weave a story from the individual quotes, the context in which these quotes came up is not really provided. What were the parts of the trainings that elicited participants to share certain experiences, or the participant engagements that elicited facilitators to consider certain ideas? 

Reviewer 2 Report

This is an important study that demonstrates the importance of continued co-design in beginning to address high prevalence of FASD in First Nations communities. It offers some preliminary ideas for adapting FASD assessment in this community, and forms the foundation for more detailed future work in this project.

My specific comments follow below:

Abstract

Line 24. Consider replacing ‘these children’ with ‘First Nations children’

Line 25. Is the word ‘role’ missing from this sentence: ‘more prominent [role?] for Aboriginal Health Workers’

Introduction

Line 43: Is the word ‘to’ missing from this sentence? ‘an important phase in assisting people [to?] gain access to effective healthcare services’

Methods

L142 – 147 duplicate phrase ‘Demographics of 147 workshop attendees are provided in Table 1.’

More details on recruitment method are required (e.g. advertising methods, potential reach, % sign up, target population)

Please report what specific diagnostic guidelines were used (Line 108)

Results:

L263 -264: There seems to be an error in the results, as the figures indicate the opposite direction to what is included in the text: Attendee confidence improved from pre-workshop (M = 2.85, SD = .88) to post-workshop (M = 2.23, SD = .71).’ I.e. the figures indicate a decline in confidence from pre- to post-workshop

The narrative synthesis results are hard to follow, due to inconsistent formatting (use of italics), which make it difficult to determine which parts are quotes, and which are interpretation

Discussion:

The discussion summarises the key points clearly and the limitations are broadly appropriate. It seems that one case study was used in the workshop ('This involved demonstration of the assessment measure, hands-on practice and scoring of a case sample. )' 

It would be useful to present details of this case sample, as this is likely to have influenced results. e.g. if the case sample was a more prototypical example of FASD (with clear facial dysmorphology, significant impairment across domains etc) then this may have been easier to score (and attendees would have reported more confidence). In future research it would be useful to include a range of case studies to reflect the diversity in presentation of the full spectrum of FASD and the difficulties that this can lead to in terms of case detection.

Round 2

Reviewer 1 Report

Thank you for the clarification regarding the qualitative work/description of yarning - as a substantial amount of qualitative research is done in more "formal" focus group settings, I think that readers unfamiliar with yarning, like myself, may be evaluating the qualitative work through this more formal lens. The only thing I would suggest adding is a comment on the importance of both collecting information and presenting it in the way that aligns with cultural practice, as it appears you do. 

Author Response

We would like to thank Reviewer 1 for their thoughtful comments as well as for the opportunity to continue to improve the clarity of our manuscript. 

Reviewer 1 comment 

Thank you for the clarification regarding the qualitative work/description of yarning - as a substantial amount of qualitative research is done in more "formal" focus group settings, I think that readers unfamiliar with yarning, like myself, may be evaluating the qualitative work through this more formal lens. The only thing I would suggest adding is a comment on the importance of both collecting information and presenting it in the way that aligns with cultural practice, as it appears you do. 

Response: 

Thank you for this feedback. We have elaborated on the use of yarning as a culturally responsive methodology on page 4.